# The Aqueous Leaf Extract of the Medicinal Herb *Costus speciosus* Suppresses Influenza A H1N1 Viral Activity under In Vitro and In Vivo Conditions

**DOI:** 10.3390/v15061375

**Published:** 2023-06-15

**Authors:** Amal Senevirathne, E. H. T. Thulshan Jayathilaka, D. K. Haluwana, Kiramage Chathuranga, Mahinda Senevirathne, Ji-Soo Jeong, Tae-Won Kim, Jong-Soo Lee, Mahanama De Zoysa

**Affiliations:** 1College of Veterinary Medicine and Research Institute of Veterinary Medicine, Chungnam National University, Yuseong-go, Daejeon 34134, Republic of Korea; amal.senevirathne@cnu.ac.kr (A.S.); thimira.thulshan@o.cnu.ac.kr (E.H.T.T.J.); dhammikaaz@gmail.com (D.K.H.); chathu0518@cnu.ac.kr (K.C.); jisooj9543@gmail.com (J.-S.J.); taewonkim@cnu.ac.kr (T.-W.K.); 2Department of Food Science and Technology, Faculty of Applied Science, Sabaragamuwa University of Sri Lanka, Belihuloya 70140, Sri Lanka; msaraths@appsc.sab.ac.lk

**Keywords:** *Costus speciosus*, leaf extract, TB100, influenza A (H1N1), A (H3N2), A(H9N2), antiviral effect, transcriptional activator

## Abstract

This study investigated the antiviral activity of aqueous leaf extract of *Costus speciosus* (TB100) against influenza A. Pretreatment of TB100 in RAW264.7 cells enhanced antiviral activity in an assay using the green fluorescence-expressing influenza A/Puerto Rico/8/1934 (H1N1) virus. The fifty percent effective concentration (EC50) and fifty percent cytotoxic concentration (CC50) were determined to be 15.19 ± 0.61 and 117.12 ± 18.31 µg/mL, respectively, for RAW264.7 cells. Based on fluorescent microscopy, green fluorescence protein (GFP) expression and viral copy number reduction confirmed that TB100 inhibited viral replication in murine RAW264.7 and human A549 and HEp2 cells. In vitro pretreatment with TB100 induced the phosphorylation of transcriptional activators TBK1, IRF3, STAT1, IKB-α, and p65 associated with interferon pathways, indicating the activation of antiviral defenses. The safety and protective efficacy of TB100 were assessed in BALB/c mice as an oral treatment and the results confirmed that it was safe and effective against influenza A/Puerto Rico/8/1934 (H1N1), A/Philippines/2/2008 (H3N2), and A/Chicken/Korea/116/2004 (H9N2). High-performance liquid chromatography of aqueous extracts led to the identification of cinnamic, caffeic, and chlorogenic acids as potential chemicals for antiviral responses. Further confirmatory studies using these acids revealed that each of them confers significant antiviral effects against influenza when used as pretreatment and enhances the antiviral response in a time-dependent manner. These findings suggest that TB100 has the potential to be developed into an antiviral agent that is effective against seasonal influenza.

## 1. Introduction

Periodic and sporadic pandemic infections caused by influenza viruses are serious global health threats despite modern medical advancements causing approximately 500,000 deaths annually worldwide, with an estimated 16.7 deaths per 100,000 people in the United States alone [1]. Influenza viruses are highly successful pathogens because they rapidly mutate into novel variants in circulation; thus, a singular immunization scheme will not curb the disease in the long run [2]. The only successful approach adopted by many countries to control their annual immunization programs is that if providing immunity against field-matched influenza strains [3]. A major concern with influenza vaccines is their inability to induce broad-spectrum immunity against multiple viral strains. The immunity generated by vaccination becomes obsolete when a novel variant evolves in circulation [4]. When the disease progresses, infected individuals in a clinical context are treated with chemical drugs that interfere with the biological processes of the virus [5]. Common examples are Rapivab (Peramivir), Relenza (Zanamivir), Tamiflu (Oseltamivir phosphate), and Xofluza (Baloxavir) [6]. Though these drugs may not provide complete protection against influenza infection, they may alleviate its severity by interfering with viral activity such as inhibiting neuraminidase and endonuclease activities [7]. Their mode of action is nonspecific and they are generally effective against both influenza A and B virus types [8]. To be proactive in disease prevention, therapeutic interventions other than immunization are essential for rapid deployment.

The natural process of recovery from a viral infection requires immunity resulting from either disease contraction or immunization. During an infection, compounds or treatments that can augment immune responses or suppress viral activity can be effective. In the search for antiviral compounds, gathering ethnomedical knowledge and analyzing herbs and their active substances enables the identification of a plethora of phytochemicals with proven medical properties [9]. However, harnessing these natural resources requires considerable effort because of the tremendous chemical and intra- and inter-species diversity among herbs. Additionally, the scarcity of valuable herbs makes it difficult to obtain them in appreciable quantities. Recently, numerous screening efforts have been undertaken worldwide to identify medicinally important herbs against human diseases [10]. Most often, the medicinal values of herbs known by traditional communities are passed down through generations, unknown to the scientific community. Hence, proper scientific investigations and documentation are essential to ensure the survival of such knowledge. Herbs are natural products in which chemicals are available at biological concentrations; hence, they can often be utilized along with conventional medicines and may be cost-effectively integrated into holistic management programs for seasonal diseases such as influenza [11].

*Costus speciosus* is a well-known medicinal herb used in traditional medicine in South Asian countries such as Sri Lanka, India, and other tropical regions. It is widely used as a treatment for diabetes [12]. This plant, popularly known as crepe ginger, spiral ginger, or insulin plant is a perennial rhizomatous herb that contains numerous medicinally important phytochemicals [13]. The known compounds of *C. speciosus* have intensive pharmacological effects owing to their antimicrobial, antioxidant, insecticidal, anticancer, and antidiabetic properties. The major active ingredients present in *C. speciosus* rhizome include diosgenin, camphene, costunolide, lupeol, zerumbone, and eremanthin, which have antiviral and antimicrobial properties [14,15]. Phytochemicals are known to distribute throughout the plant such as, leaves, stem and most notably, the rhizome. Most studies have focused on rhyzomes, but not the leaf material [13]. Traditionally, the leaves are consumed in raw or minimally processed form and can regulate blood glucose levels significantly. We speculated that the leaf material may contain antiviral compounds that are yet to be identified, and we tested this material for its effectiveness against influenza infections. Therefore, we aimed to evaluate the effectiveness of the *C. speciosus* leaf extract (TB100) against influenza A/Puerto Rico/8/1934 (H1N1) as a model system using in vitro and in vivo strategies.

The results of this study indicated that the *C. speciosus* aqueous extract (TB100) displays appreciable antiviral activity against influenza A/Puerto Rico/8/1934 (H1N1), A/Philippines/2/2008 (H3N2), and A/Chicken/Korea/116/2004 (H9N2) and contains active antiviral compounds such as cinnamic, caffeic, and chlorogenic acids. Here, we demonstrated that selected active compounds have immunostimulatory properties. The leaf extract and its compounds may be developed as an oral treatment strategy to prevent seasonal influenza. Further investigations are warranted to determine the broadness of the antiviral activity against other important human and animal viruses while utilizing the active compounds present in this herb to fabricate novel drug candidates.

## 2. Materials and Methods

### 2.1. Preparation of C. speciosus TB100 Aqueous Extract

The identification and authentication of the plant species were conducted by the Faculty of Applied Sciences, Sabaragamuwa University of Sri Lanka. The leaves of *C. speciosus* were collected, washed with clean water, and then drained and desiccated for 3 d at 50 °C. The dried leaf material (260 g) was ground into a fine powder for efficient aqueous extraction [16] and subjected to hot water and methanol extraction. Boiling water (2 L) was used to extract 130 g of powder at 100 °C for 150 min, repeated thrice to prepare the hot water extract. The pH was adjusted to 7.4 pH and filtered through 0.45 μm filter units (Acrodisc; Pall Corporation, Washington, NY, USA). The filtered samples were freeze-dried and aliquots were prepared at a concentration of 1 mg/mL, adjusted pH of 7.4, and filtered through 0.2 μm filters for cell culture experiments. For methanol extraction, 80% methanol was added into leaf materials in an Erlenmeyer flask and soaked for three days at room temperature. Extraction of methanol dissolved substances was performed by evaporating the methanol extract. Both aqueous and methanol extracts were subjected to high-performance liquid chromatography (HPLC) identification of the active compounds.

### 2.2. Removal of Endotoxin Contaminants

Endotoxin contaminants were eliminated via filtration through Acrodisc Units with a Mustang E Membrane; a 0.2 μm pore size (MSTG25E3; Pall Cooperation, Washington, NY, USA) at a flow rate of 1 mL/min. The resulting filtrates were examined for residual endotoxin levels using a commercial chromogenic endotoxin determination kit (Thermo Fisher, Waltham, MA, USA). Endotoxin contamination was determined against the standard calibration curve provided with the kit, following the manufacturer’s protocol.

### 2.3. Cell Culture and In Vitro Infection of A/Puerto Rico/8/1934 (H1N1)

Cultures of the murine macrophage cell line RAW264.7, human laryngeal carcinoma cell line HEp2 (ATCC CCL-23, Manassas, VA, USA), and lung adenocarcinoma A549 cells (ATCC CCL-185, Manassas, VA, USA) were routinely prepared in Dulbecco’s Modified Eagle’s medium (DMEM; Lonza, Basel, Switzerland) supplemented with a 10% fetal bovine serum (FBS; Serana, CT, USA) and a 1% broad spectrum antibiotic (Sigma; St. Louis, MO, USA) at 37 °C in a 5% CO_2_ atmosphere. For the viral infection, the green fluorescent protein (GFP)-expressing influenza strain, influenza A/Puerto Rico/8/1934 (H1N1) (A H1N1 PR8-GFP) was used. Viral infection and titer determination were conducted using standard TCID50 determination assay [17,18]. Serum-free DMEM was utilized as virus infection medium during in vitro studies.

### 2.4. Determination of 50% Cytotoxic Concentration (CC50) of TB100

The cytotoxicity of TB100 was determined in RAW264.7, A549, and HEp2 cells. Cell lines were seeded in 96-well plates in replicates (two wells per treatment) at a seeding density of 1 × 10^4^ cells/well in DMEM complete medium (10% FBS, 1% antibiotics) and incubated for 12 h in a 5% CO_2_ atmosphere at 37 °C. Cells were treated with TB100, prepared in a complete medium consisting of 0, 1.25, 2.5, 5, 10, 25, 50, 100, 150, 200, and 300 μg/mL. The culture plates were gently shaken to ensure proper distribution and then returned to the incubator for 24 h. Subsequently, the number of live cells was determined using the (3-(4,5-dimethylthiazole-2-yl)-2.5-diphenyltetrazolium bromide (MTT) assay [19,20]. Briefly, the cell supernatant was aspirated and MTT reagent (100 μL) was added to each well, and the cells were incubated for five hours at 37 °C. After incubation, DMSO (50 μL) was added to dissolve the generated formazan. Colorimetric changes were quantified by measuring the absorbance at 595 nm. Using the absorbance values, a 50% cytotoxic concentration (CC50) was determined following the standard statistical method relied on non-linear regression model [21]. Results were obtained by three independent experiments.

### 2.5. Determination of 50% Effective Concentration (EC50) of TB100

The 50% percent effective concentration of TB100 was determined using the RAW264.7, A549, and HEp2 cell lines. The cells were grown in 12-well plates in replicates (two wells per treatment) at a seeding density of 1 × 10^5^ cells/well in the complete medium (DMEM, 10% FBS, and 1% antibiotics at 37 °C in a 5% CO_2_ atmosphere). After 12 h, cells were treated with variable concentrations at 0, 5, 10, 15, 20, 40, and 80 μg/mL of TB100, prepared in a complete medium. Cells treated with the medium alone were used as negative controls. Cell culture plates were gently swirled and returned to the incubator. After incubation for 12 h, the culture supernatant was aspirated and the cells were washed thrice with DMEM. Subsequently, the cells were infected with the influenza A (H1N1) PR8-GFP virus at 0.1 MOI for RAW264.7 and 1.0 MOI for A549 and HEp2 cell lines. After 2 h of incubation, the cells were replenished with the complete medium and further incubated for 24 h. GFP expression was observed under a fluorescence microscope with appropriate filter settings (Leica, Lahn-Dill-Kreis, Wetzlar, Germany), and images were captured under green fluorescence and bright field conditions. The cell supernatants were removed, and the whole monolayers of two wells were combined and used to isolate total RNA using a commercial kit (Nucleospin; Machery Nagel, GmbH, Duren, Germany). After RNA quantification, 1 μg of RNA from each sample was converted to cDNA using the Takara PrimeScript 1st strand cDNA synthesis kit following the manufacturer’s protocol (Takara Bio, Tokyo, Japan). The elution of cDNA was performed in 20 µL of nuclease-free water and subsequently diluted into 730 μL of nuclease-free water. To determine the viral copy number, the GFP open reading frame (ORF) of the virus was used as a calibrator gene and cloned into a plasmid. A calibration curve was developed between logs 8.6 and 1.6, plasmid copy numbers and CT values. Intracellular viral copy numbers were determined at the reference curve based on the CT values obtained for virus-infected samples [22]. Results were obtained by three independent experiments.

### 2.6. Preliminary Assessment of TB100 Time of Addition Assay

TB100 time of addition on the antiviral effect was evaluated using RAW264.7 cells (Appendix A). RAW264.7 cells were seeded in 12-well plates in replicate at a seeding density of 2.5 × 10^5^ cells/well in complete medium in replicates (two wells per treatment). Cells were incubated for 12 h in a 5% CO_2_ atmosphere at 37 °C. To evaluate the pretreatment effect, TB100 was added to the culture medium at 12, 8, 4, and 2 h before the time of virus addition for infection. TB100-treated (20 µg/mL) cells were washed with medium alone three times. A (H1N1) PR8-GFP virus was added at a multiplicity of infection (MOI) 0.1 prepared in 500 µL/well of serum-free medium. Plates were firmly swirled for even distribution and further incubated for 2 h for infection. After incubation, the infection medium was replaced by the DMEM complete medium and sent back to incubation for 24 h. At the end of the incubation, the level of infection was evaluated by fluorescence microscopy. After imaging, cells were detached by trypsinization and collected by centrifugation (500× *g* for 10 min). Cells were resuspended in 500 µL of PBS and the GFP absorbance was measured by a fluorescence spectrophotometer (GloMax Promega; Madison, WI, USA). TB100 post-treatment effect was evaluated in the same manner of cell seeding and incubation for 12 h where the infection was conducted. TB100 (20 µg/mL) was added at 2, 4, 8, and 12 h after infection. Cells were further incubated for 24 h, and fluorescent imaging and GFP absorbance quantification were conducted. Experiments were conducted in three independent trials for the activity confirmation. Results were obtained by three independent experiments.

### 2.7. Preliminary Assessement of TB100 Virucidal, Attachment, and Entry Assay

The virucidal activity of TB100 was assessed in RAW264.7 cells. Cells were propagated in 12-well plates in replicates (two wells per treatment) and incubated for 12 h. TB100, 20 µg/mL concentration, was mixed into A (H1N1) PR8-GFP at 0.1 MOI in a 500 µL culture volume and incubated for 20 min at 37 °C (Appendix A). The co-mix was then added to the pre-washed (three times with DMEM) cell monolayer and swirled to equal spread. Infection was conducted for 2 h in serum-free culture medium and then the infection medium was removed and a fresh complete medium was added. Infection was continued for 24 h; the level of infection was evaluated by fluorescence microscopy and GFP absorbance measurements. For the TB100 effect on viral attachment, RAW264.7 cells were incubated with a co-mix of TB100 and A (H1N1) PR8-GFP similar to a previous study; however, it was incubated for 2 h at 4 °C for infection, and the medium was replaced with DMEM complete media (Appendix A). Viral infection was evaluated by fluorescent imaging and GFP measurements. The TB100 effect on viral entry was evaluated in RAW264.7 cells. Here, first, the viral infection was carried out for 2 h in a similar aforementioned manner, and the TB100 treatment was initiated 2 h after the viral infection (Appendix A). Assessment of viral infection was conducted 24 h after the incubation by fluorescent microscopy and GFP absorbance measurements. Results were obtained by three independent experiments.

### 2.8. Assessment of TB100 Antiviral Activity

Murine macrophage RAW264.7 cell lines were seeded in 12-well plates in replicate (two wells per treatment) at a seeding density of 2.5 × 10^5^ cells/well in DMEM, 10% FBS, and 1% broad-spectrum antibiotics (Sigma, St. Louis, MO, USA). Cells were incubated at 37 °C for 12 h in a 5% CO_2_ atmosphere. Subsequently, the cells were washed thrice with DMEM and treated with variable concentrations of TB100 prepared in a complete medium. Based on cytotoxicity data, noncytotoxic concentrations such as 5, 10, 20, and 40 μg/ mL were used for antiviral assays. The cells were co-incubated with the TB100 herbal extract for 12 h. As a positive control (PC), cells were treated with 50 and 100 units of interferon (IFN)-β, while the medium alone (MO) was treated as the negative control. After incubation, cell monolayers were washed thrice with DMEM and infected with influenza A (H1N1) PR8–GFP at a MOI 0.1, prepared in a 500 µL culture medium. The infection process proceeded for 2 h with intermittent slow shaking to facilitate viral adhesion. Subsequently, virus-containing supernatant was aspirated, and cells were replenished with a complete medium and further incubated for 24 h in a 5% CO_2_ atmosphere at 37 °C. After incubation, the cells were observed under a fluorescence microscope, and images were taken with appropriate filter settings for comparison. After imaging, cell monolayers were detached by trypsinization, and cell viability was determined using the standard trypan blue exclusion method [23]. The remaining cell suspension was used to determine GFP absorbance using a fluorescence spectrophotometer (GloMax Promega; Madison, WI, USA). Initial confirmation of antiviral activity was performed for both aqueous and methanol-extracted TB100 samples. Three independent experiments were conducted.

### 2.9. TB100 Mediated Induction of IFN-β and IL6

RAW264.7 cells were treated in replicates (two wells per treatment) with TB100 at 20 and 40 µg/mL and incubated at 37 °C in a 5% CO_2_ atmosphere. Culture supernatants were collected at 12 and 24 h intervals. Secreted amounts of IFN-β and IL6 were assessed using a commercial Enzyme-Linked Immunosorbent Assay (ELISA) kit (BD Biosciences, Franklin Lakes, NJ, USA). As a positive control, RAW264.7 cells were treated with 10 ng/mL lipopolysaccharide (*Salmonella* LPS; Sigma). ELISA was conducted according to the manufacturer’s recommendations. Furthermore, the expression of IFN-β and IL6 was confirmed at the mRNA level by qRT-PCR. Briefly, RAW264.7 cells were cultured in 6-well plates in replicates (two wells per treatment) at a seeding density of 3.0 × 10^5^ cells/well and then were incubated for 12 h at 37 °C in a 5% CO_2_ atmosphere. After incubation, TB100 was treated at 20 and 40 μg/mL concentrations. Cells treated with the MO or with 50 units of IFN-β (Sigma) were used as negative and positive controls, respectively. After 24 h of incubation, the cells were lysed, total RNA was extracted, and cDNA was synthesized, as previously described. The induction of cytokine genes by TB100 was measured by qRT-PCR using SYBR Green PCR Master Mix (Takara, Tokyo, Japan). qRT-PCR was performed using the Thermal Cycle Dice Real Time System II (Takara, Tokyo, Japan). Expression levels of each marker gene were presented as a 2^−ΔΔCT^ fold-change according to a previously described method [24]. The primers used for qRT-PCR are listed in Table 1. Three independent experiments were conducted.

### 2.10. Western Blot Detection of Transcriptional Regulator Phosphorylation

Activation of transcriptional regulators—tank binding kinase 1 (TBK1), interferon regulatory factor 3 (IRF3), signal transducer and activator of transcription 1 (STAT1), nuclear activator of kappa light polypeptide gene enhancer in B-cell inhibitor alpha (IKB-α), and p65 by TB100—was assessed in the RAW264.7 cell line. RAW264.7 cells were seeded in 12-well plates (two wells per treatment) and treated with TB100 (40 µg/mL). IFN-β (50 units) was used as the positive control and the MO was considered as the negative control. Cells were incubated at 37 °C in a 5% CO_2_ atmosphere and collected for protein isolation at 0, 8, 12, and 24 h after treatment. Cell suspension was centrifuged at 500× *g* at 4 °C, the supernatant was removed and the pellet was collected for protein extraction. Cell lysis was conducted in the ProEx CETi Lysis Buffer (Trans Lab, Daejeon, Korea) using 300 µL per pellet. Cell suspension was briefly sonicated at a 20% amplitude for 1 min (Sonic Vibra Cell, Newton, CT, USA), centrifuged at high speed (>10,000× *g*), and the supernatants were collected and stored at −80 °C for future use. Proteins were quantified using the standard Bradford method [28]. Purified proteins (25 µg per well) were resolved in sodium dodecyl sulfate–polyacrylamide gel electrophoresis (SDS-PAGE). The resolved proteins were transferred onto polyvinylidene difluoride membranes (Immobilon, Kenilworth, NJ, USA). Western blot analysis was conducted with the following Rabbit anti-mouse monoclonal antibodies (mAb): IKB-α; cat: 9242S, p-IKB-α; cat: 2859S, p65; cat: 4764s, p-p65; cat:3033S, IRF3; cat: D83B9, p-IRF3: cat: 4947S, STAT1; cat: 9172S, p-STAT1; cat: 9167S, TBK-1; cat: 3504S, p-TBK-1; cat: 5483S, S (Cell Signaling, Danvers, MA, USA). Anti-rabbit horseradish peroxidase (HRP)-tagged IgG antibody (cat: 7074; Cell Signaling Technology, Danvers, MA, USA) was used as the secondary antibody at a 1:3000 dilution. Membrane washing was performed three times (10 min each) with a washing buffer (PBS—0.05% Tween 20, 5% BSA). For detection, a chemiluminescence assay was performed using a commercial kit (Western Femto ECL kit, LPS Solution, Daejeon, Korea). Images were visualized using a chemiluminescence detection system (Fusion Solo, Vilber, Lourmat, France). Quantification of band intensity was conducted using Image J (Fiji) software (V 1.54) (National Institute of Health, Rockville Pike, Bethesda, MD, USA) [29].

### 2.11. TB100 Antiviral Activity—In Vivo Dose Optimization

Ethical approval to conduct the in vivo studies using BALB/c mice was obtained by the Animal Ethic Committee at Chungnam National University (202203A-CNU-18 and 202212A-CNU-252). Specific pathogen-free (SPF) female BALB/c mice (30 at five weeks old) were purchased from Orient Bio (Jungwon-gu, Seongnam-si, Gyeonggi-do, South Korea). The mice were housed in an animal facility at the Research Institute of Veterinary Medicine, Chungnam National University for 5 days for acclimatization. Purified air, antibiotic-free water, and food were provided ad libitum. After 5 days, the mice were randomly divided into six groups: group 1, TB100-0.1 mg/mL (1.05 mg/kg); group 2, TB100-0.2 mg/mL (2.10 mg/kg); group 3, TB100-0.4 mg/mL (4.20 mg/kg); group 4, TB100-0.8 mg/mL (8.4 mg/kg); group 5, PBS; and group 6, naïve (healthy control) (N = 30, *n* = 5) (Appendix A). Mice were orally treated with TB100 in the above concentrations in a total volume of 200 µL of PBS on the 5th, 7th, 9th, and 11th day, and on the 12th day, they were nasally challenged with double LD50 of the A/Puerto Rico/8/1934 (H1N1) influenza strain (Appendix A). Body weight was measured daily from day one until the end of the experiment. On the 3rd day post-infection (DPI), one mouse from each group was randomly selected and euthanized to collect lung tissue for histopathological examination using hematoxylin and eosin (H&E) staining [30]. Viral titers were determined in the lung tissue using the TCID50 method [31]. At 10^th^ DPI, all remaining mice were sacrificed, and lung tissue was collected for histopathological analysis and viral titer determination. Afterward, the mice with critically low body weight (<20%) compared to initial body weight were considered as mice prone to death.

### 2.12. TB100 Antiviral Activity—In Vivo Challenge Study

The antiviral effects of TB100 against multiple influenza A strains, A/Puerto Rico/8/1934 (H1N1), A/Philipines/2/2008 (H3N2), and A/Chicken/Korea/116/2004 (H9N2) were investigated in BALB/c mice [32,33]. Antibiotic-free water and food were provided ad libitum. Female SPF BALB/c mice (80 at five weeks old) were purchased from the Hanil Laboratory Animal Center (Jeonju, Korea) and divided into 10 separate groups as follows: naïve (healthy control), PBS—H1N1, TB100—H1N1, PC—H1N1, PBS—H3N2, TB100—H3N2, PC—AH3N2, PBS—H9N1, TB100—H9N2, and PC—H9N2 (*n* = 8) (Appendix A). TB100 was fed orally according to the former schedule and optimized dose conditions at 0.8 mg/mL (8.4 mg/kg) (Appendix A). Interferon β, 1000 units per mouse, was nasally inoculated as a positive control. Upon TB100 treatment, the mice were challenged with A (H1N1), A (H3N2), and A (H9N2) strains at double LD50. Post-challenge body weight measurements were performed daily. Mice were monitored daily for mortality.

### 2.13. HPLC Analysis of Chemical Compounds Present in TB100 Aqueous, and Methanol Extracts

Liquid chromatography was performed using an Agilent 1260 high-performance liquid chromatography (HPLC) system (Agilent Technologies, Santa Clara, CA, USA) equipped with an Eclipse Plus C18 column (part no. 959763-902, lot no. B21098, 3.5 µm, 2.1 × 150 mm; Agilent) at 40 °C. The mobile phases were 0.1% formic acid in distilled water and acetonitrile, with the gradient conditions set at a flow rate of 0.25 mL/min as follows: 3–15% B, 0–2; 15–50% B, 2–13; 100–100% B, 20–23; 100-3% B, 23–23.5; and 3–3% B, 23.5–28. The injection volume was 3 µL and the UV detector was set at 280 nm. Chemical standards for gallic, chlorogenic, caffeic, ferulic, and cinnamic acids and catechin, as well as quercetin were quantified in TB100. Based on the literature, cinnamic, caffeic, and chlorogenic acids were selected for further analysis [34].

### 2.14. Antiviral Activity of Active Compounds

Based on the HPLC analysis cinnamic, cafffeic, and chromogenic acids were identified as three potential antiviral chemicals in the TB100 methanol and water extracts. They were purchased from Sigma (St. Louis, MO, USA) and prepared a 2 M stock solution: subsequent dilutions were prepared in a cell culture medium for antiviral assays. In vitro, the antiviral assay was conducted using 0, 20, 40, and 55 μM concentrations on RAW264.7 cells. Cells were pretreated with each compound at the indicated concentrations for 12 h and infected with the influenza A (H1N1) PR8-GFP virus. Fluorescence microscopy and GFP absorbance measurements were conducted as described in the Methods section. The activation of protein transcriptional regulators associated with the interferon pathway was investigated by Western blot analysis and qRT-PCR following a procedure similar to that used for Western blotting and qRT-PCR. Afterward, the RAW264.7 cells were treated with 2 mM of each of the selected compounds to obtain the increased response of proteins. The EC50 of each compound was determined based on the GFP absorbance measurements. The CC50 for each compound was determined at concentrations between 0 and 10 mM. Briefly, RAW264.7 cells were seeded in 96-well plate (three wells per treatment) and incubated for 12 h. Cells were treated with a medium containing 0, 1, 2, 5 and 10 mM cinnamic, caffeic and chlorogenic acid compounds. Cells were further incubated for 24 h, and the level of cytotoxicity was assessed using a commercial kit (EZ-Cytox, Geumcheon-gu, Seoul, Korea) according to the manufacturer’s protocol [35].

### 2.15. Statistical Analysis

All statistical analyses were conducted using GraphPad Prism software (San Diego, CA, USA). Data are presented as means of each experiment with ±standard deviation (SD). Treatment means were compared using analysis of variance (ANOVA) with multiple comparison methods. An unpaired t-test was performed at the end of the body weight measurements to compare the body weight changes after treatment with those of the PBS control. Significant levels of the Kaplan–Meier survival curves were determined using the log-rank (Mantel–Cox) test. Mean differences were considered significant at *p* < 0.05.

## 3. Results

### 3.1. Determination of the CC50, EC50 and Selectivity Index of TB100

As presented in Table 2, the CC50 concentrations for RAW264.7 cells were 117.12 ± 18.31 µg/mL, and those for A549 cells were 71.86 ± 8.91 µg/mL. Notably, HEp2 cells did not show cytotoxicity until the upper limit of the test concentrations (300 µg/mL). The 50% effective concentration (EC50) values for TB100 on the RAW264.7, A549, and HEp2 cells were 15.19 ± 0.61, 13.64 ± 0.55, and 16.78 ± 0.42 µg/mL, respectively (Table 2). EC50 values were estimated based on the CT values obtained from qRT-PCR (Appendix A). It was also evident that the A549 and HEp2 cells resulted in a comparatively lower infection than that of RAW264.7 cells (Appendix A). The selected treatment concentrations of TB100, 20, and 40 µg/mL did not cause significant cytotoxicity compared to the MO group (Appendix A). The VO group resulted significant cell death owing to viral activity. Viability results obtained for both 20 and 40 µg/mL concentrations were comparable to each other. The selectivity indices of TB100 for RAW264.7 and A549 cells were estimated to be 7.68 ± 0.89 and 5.25 ± 0.44, respectively. Because the cytotoxicity of TB100 in the HEp2 cells was above 300 µg/mL, the selectivity index was higher than 17.87.

### 3.2. TB100 Effect on Time of Addition, Virucidal, Attachment, and Entry of A/Puerto Rico/8/1934(H1N1) Influenza

An initial antiviral assessment of *C. speciosus* was conducted for both aqueous and methanol-extracted samples (unpublished data). Because we did not observe a significant difference in viral suppression between the two extracts, the rest of the analysis was conducted using aqueous extracted samples, hereon referred to as TB100. Aqueous extraction of *C. speciosus* could produce approximately 15% of the *w*/*v* dried mass. After filtration and pH adjustment, TB100 was assessed for its cytotoxicity and antiviral activity against the GFP-expressing influenza A/Puerto Rico/8/1934 (H1N1) (A (H1N1) PR8-GFP) strain using in vitro cell culture model. Owing to the ability of the virus to express GFP during infection, viral activity can be directly observed based on a quantitative assessment of GFP expression. Before the antiviral assessment, TB100 was subjected to endotoxin removal, and endotoxicity was lowered to levels acceptable for in vivo utilization based on recommendations by the Food and Drug Administration (FDA; USA guidelines) (unpublished data). Based on the cytotoxicity assessment, we chose the noncytotoxic concentrations of 20 and 40 μg/mL to determine the antiviral activity of TB100. As a preliminary investigation to identify the activity of TB100 on viral replication, a time of addition experiment was conducted on RAW264.7 cells. TB100 was evaluated as a pretreatment candidate by adding it at various time points before the infection. The addition of TB100 12, 8, 4, and 2 h before the infection revealed a significant viral suppression ability. A significantly lower replication was observed at 12 h of treatment (*p* < 0.0001) compared to the 2 h time point (Figure 1A,a1). However, we could not observe the TB100 effect on A (H1N1) suppression as either after treatment or before treatment. We observed that a delay in time of addition of TB100 into the infected cells significantly reduced the efficacy of TB100 treatment on viral suppression. Here, the maximum suppression of viral activity was observed when TB100 was added as early as 2 h after the infection (*p* < 0.0001) compared to 12 h after incubation (Figure 1B,b1). Furthermore, the TB100 effects on virucidal activity (Figure 1C,c1), viral attachment (Figure 1D,d1), and entry (Figure 1E,e1) were also investigated, which resulted in no significant effect on the suppression of viral entry and initiation of replication. Due to the profound pretreatment effect of TB100, subsequent studies were focused on the pretreatment antiviral effect.

### 3.3. TB100 Possesses Pretreatment Antiviral Properties against A/Puerto Rico/8/1934 (H1N1) Influenza

The pretreatment effect of TB100 on the murine macrophage RAW264.7 cell line resulted in significant, dose-dependent antiviral activity against influenza A (H1N1). Comparing cell treatment concentration at 20 and 40 µg/mL, the TB100 results at 40 µg/mL were comparable to those of IFN-β (PC). The results were compared with those of the MO (non-treated), VO, and positive controls. A preexposure of 12 h to TB100 at 20 and 40 µg/mL could significantly reduce intracellular viral replication (Figure 2A,B). To obtain quantitative measurements of viral reduction, the cells were detached from the culture plate and prepared in a suspension form to measure the absorbance (Figure 2B). Cells infected with A (H1N1) PR8-GFP without a treatment (VO) group demonstrated heavy infection (*p* < 0.0001) compared to the MO group. Furthermore, results indicated that IFN-β (50 units) and TB100 concentrations, at both 20 and 40 µg/mL, were significantly lower than those of the VO control based on green fluorescence emission. Importantly, none of the treatment concentrations caused a significant cytotoxic effect on RAW264.7, ensuring that the TB100 extract is safe for cultured cells in vitro (Figure 2C). At the same 20 and 40 µg/mL concentrations, an approximately four-log reduction in viral copy number was observed for the TB100-treated sample, which was comparable to that of the IFN-β-treated positive control (*p* < 0.0001) (Figure 2D). The viral copy number was estimated based on GFP ORF as a calibrator gene. The GFP ORF was cloned into a plasmid vector (a modified pGFPuv vector) and used to develop a standard curve between the plasmid copy number and the CT value. Viral copy numbers were estimated based on a standard curve using the CT values obtained for each sample. 

### 3.4. Assessment for Induction of IFN-β, IL6 and Transcriptional Activators by TB100

The effect of TB100 on IFN-β and IL6 was evaluated on RAW264.7 cells using ELISA and qRT-PCR analysis. For the ELISA, RAW264.7 cells were treated with TB100 at 20 and 40 µg/mL and compared against the MO (negative control) and the *Salmonella* LPS (10 ng/mL) positive control. Here, IFN-β was not used, as it was a target cytokine in the ELISA measurements. After treatment, culture supernatants were collected at 12 and 24 h intervals. A significant level of IFN-β secretion for TB100-treated cells (*p* = 0.0004) was observed at 24 h after treatment at 40 µg/mL (Figure 3A(a)), whereas the positive control showed significant induction of IFN-β at 12 h after treatment (*p* = 0.0428). A significant level of IL6 secretion was observed at 12 h after incubation at 40 µg/mL concentration for the TB100-treated samples (*p* = 0.0376) (Figure 3A(a)) and a more profound response was observed at the 24 h time point for both 20 and 40 µg/mL treatment concentrations (*p* < 0.0001). Confirmation of IFN-β and IL6 induction by TB100 was also delivered by qRT-PCR, which corroborated the ELISA results for both markers at 24 h after treatment (Figure 3A(b)).

Activation of antiviral marker proteins due to TB100 pretreatment was observed in RAW264.7 cells. Among the selected protein markers of the interferon pathway, TBK1, IRF3, STAT1, IKB-α, and p65 were seen phosphorylated owing to TB100 exposure. Compared to IFN-β (PC), the responses generated by TB100 peaked at early time points (8 h) and showed a decreasing trend towards the 24 h time point. However, IFN-β exposure caused a perpetual phosphorylate response in treated RAW264.7 cells (Figure 3B(a)). Quantification of band intensity (Appendix A) demonstrates the highest level of phosphorylation of TBK1 and STAT1 at 8 h after treatment, whereas IRF3 and p65 phosphorylation was high at 12 h after treatment. IKB-α phosphorylation was highest at 8 h after treatment and slightly reduced at 24 h after treatment. Apart from the level of phosphorylation, TB100 affects the levels of mRNA expression of some of the markers such as IKB-α IRF3, STAT1, etc. (Figure 3B(b)). Observations made by Western blot analysis and the protein expression patterns observed in qRT-PCR corroborated each other, indicating that TB100 could activate antiviral defense in RAW264.7 cells by activating interferon pathways potentially via multiple mechanisms. It is important to note that the effect of TB100 may be governed by multiple active substances present in the aqueous leaf extract, which was not investigated here.

### 3.5. TB100 In Vivo Dose Optimization and Antiviral Activity

TB100 oral inoculation and dose optimization were performed using the BALB/c mice model. TB100 was prepared at concentrations of 0.1 (1.05 mg/kg), 0.2 (2.10 mg/kg), 0.4 (4.20 mg/kg), and 0.8 mg/mL (8.40 mg/kg). The PBS and naïve controls were used for comparison. On the fifth, seventh, ninth, and eleventh day, the TB100 formulations were administered to each group (200 µL total volume) via oral gavage (Appendix A). We observed that oral treatment of TB100 in mice did not cause any observable health deterioration or body weight loss during the feeding period (Figure 4A(a)). Their behavior and appetite were comparable to those in the naïve and PBS groups. Up on the influenza A/Puerto Rico/8/1934 (H1N1) challenge on the 12th day, the subsequent changes in body weight resembled the relative protection caused by the treatment (Figure 4A(a)). Toward the end of the experiment, mice in the VO control group were severely affected by the influenza challenge resulting in a 100% mortality, whereas the TB100-treated mice demonstrated a dose-dependent protection with zero mortality. For quantitative and qualitative comparisons of protection under each treatment condition, histopathological examination of the lung tissue and viral titer determination were conducted at 3^rd^ and 10^th^ DPI. Severe signs of inflammation, immune cell infiltration exudate-filled tissue cavities, and fluids were evident in the H&E-stained lung tissue images of the VO group (Figure 4A(b). Histopathological observations revealed a dose-dependent protection with a clear and near-normal lung tissue architecture observed in the TB100 0.8 mg/mL group. The lung tissue titer estimation at 3^rd^ and 10^th^ DPI revealed a significant >3-log reduction in TCID50 values for the TB100 0.8 mg/mL group (*p* = 0.0002) by 10^th^ DPI. By evaluating the body weight change, histopathological examination, and reduction in viral load, we decided that TB100 at 0.8 mg/mL (8.40 mg/kg) would be suitable for in vivo inoculation studies to obtain the maximum anticipated protection.

### 3.6. In Vivo Challenge Study

The antiviral effects of TB100 on other types of A influenza viruses were evaluated using a BALB/c mice model. The inoculation scheme was similar to that used for the dose optimization schedule (Appendix A). After four consecutive oral treatments, the mice were nasally challenged with influenza A/Puerto Rico/8/1934 (H1N1), A/Philipines/2/2008 (H3N2), and A/Chicken/Korea/116/2004 (H9N2) viral strains using a double LD50. Body weight measurements indicated that the non-inoculated mice (the H1N1-PBS group) were highly susceptible to the A (H1N1) challenge, with the earliest onset of mortality at 6^th^ DPI. By 8^th^ DPI, a 100% mortality was recorded, while the TB100-treated mice could significantly tolerate the infection with delayed onset of mortality (9^th^ DPI) and, with an only 50% mortality recorded by 10^th^ DPI. In PC, an only 25% mortality was recorded by 8^th^ DPI (Figure 4B(a1,b1)). The average mice body weight was observed less than 80 % of the initial body weight before reaching 7^th^ DPI. Regarding the A (H3N2) and A (H9N2) challenge, the mice demonstrated considerable tolerance against infection; however, the highest mortality was recorded in non-treated mice (H3N2-PBS and H9N2-PBS groups). In the H3N2-PBS group, a 62.5% mortality was observed between 6^th^ and 9^th^ DPI, whereas the TB100-treated group resulted in a 50% mortality on 9^th^ DPI. Moreover, the PC also recorded a 50% mortality earlier than TB100 that was between 7^th^ and 9^th^ DPI. The body weight changes also demonstrated a declining body weight for both PC- and TB100-treated mice (Figure 4B(a2,b2)). The challenge against A/Puerto Rico/8/34 (H9N2) demonstrated a 75% mice mortality between 6^th^ and 10^th^ DPI for the PBS-treated group, whereas the PC-treated group resulted in a 62.5% mortality by 9^th^ DPI. Highest level of protection was recorded in the TB100-treated group with a 50% recorded mortality by 9^th^ DPI. A decline in body weights was observed for both PC- and TB100-treated mice toward the end of the experiment (Figure 4B(a3,b3)). Comparing the body weight reduction pattern, time of first mortality and the total number of protected mice, TB100 resulted in a highest protective response against A/Puerto Rico/8/1934 (H1N1) followed by A/Philipines/2/2008 (H3N2) and A/Chicken/Korea/116/2004 (H9N2).

### 3.7. Identification ofTB100 Active Compounds and Their Antiviral Effects

HPLC was conducted to identify the potential antiviral compounds present in TB100. Assessments were conducted using distilled water and methanol-extracted TB100 samples. Among the array of predicted compounds, three candidates, namely cinnamic, caffeic, and chlorogenic acids, were selected as potential chemical compounds with antiviral activities against influenza A (H1N1) (Figure 5A). The respective compounds were present in both distilled water and methanol extracts. These compounds have proven efficacy against other viral species such as cinnamic acid against hepatitis C and the Zika virus [36,37], caffeic acid against influenza replication [38], and chlorogenic acid against influenza A (H1N1) and HIV [39,40]. Suppression of viral replication was evident in GFP fluorescence microscopy, where chlorogenic and caffeic acids demonstrated a higher level of A (H1N1) inhibition than cinnamic acid (Figure 5B). Western blot analysis for phosphorylation of transcriptional activators associated with interferon pathway demonstrated phosphorylation of TBK1, IRF3, STAT1, IKB-α, and p65 by all three compounds (Figure 5C). A time-dependent increase in phosphorylation was also evident, similar to the IFN-β-positive control. The intensity of bands indicated higher levels of protein phosphorylation for markers such as IRF3, STAT1, and IKB-α. Chlorogenic acid caused time-dependent suppression of IRF3 expression (Figure 5C and Appendix A). qRT-PCR also demonstrated the induction of key antiviral markers including TBK1, IRF3, STAT1, IKB-α, OAS, and IFN-α (Figure 5D). Interestingly, all three compounds caused IL-1β suppression, and no significant change was observed against ISG56 (Figure 5D). Collectively, these compounds may contribute to the antiviral activity of TB100. The pretreatment effects of compounds such as cinnamic, caffeic, and chlorogenic acids against seasonal influenza have not been reported. Assessment of the antiviral activity of pre-exposed RAW264.7 cells to varying concentrations of the three selected compounds revealed that they significantly suppressed the intracellular replication of influenza A (H1N1). Collectively, these results suggest that the selected compounds may contribute to the antiviral properties of TB100.

### 3.8. Selectivity Index (SI) of Cinnamic, Caffeic, and Chlorogenic Acids

Fifty percent cytotoxicity and effective concentrations of cinnamic, caffeic, and chlorogenic acids were determined using in vitro assays on RAW264.7 cells. Among the three compounds, the cytotoxicity of caffeic acid was the lowest with CC50 at 8.925 mM, whereas the CC50 values of cinnamic and chlorogenic acids were 6.076 and 4.875 mM, respectively (Table 3 and Appendix A). The highest effectiveness related to the antiviral effect could be seen in chlorogenic acid (15.62 µM), whereas caffeic and cinnamic acid resulted in almost comparable outcomes at 28.18 µM and 28.64 µM, respectively (Table 3). The relatively high selectivity index of these compounds against influenza viruses demonstrates their potential as effective antiviral agents under biological conditions.

## 4. Discussion

As is common for many herbal species discovered in traditional medicine [41], *C*. *speciosus* requires more vigorous systematic studies to uncover its medicinal usage for various diseases. Here, we experimented with its aqueous leaf extract against influenza A/Puerto Rico/8/1934 (H1N1) using in vitro and in vivo assays. Furthermore, we tested the protective efficacy of these extracts against A/Philippines/2/2008 (H3N2) and A/Chicken/Korea/116/2004 (H9N2) to investigate whether it is effective against other influenza species belonging to influenza type A. Curated as a medicinal herb for centuries, this plant has not been reported to have adverse effects in medicinal applications and is safe for human use. Furthermore, we observed that the cytotoxic range of TB100 was well above the concentration required for significant inhibition of influenza viral activity. Chemical compounds such as gallic, chlorogenic, caffeic, and cinnamic acid have been reported in *C. speciosus* samples [34]. Owing to its glucose regulatory potential and antimicrobial properties, the immune-inducing capabilities of the active compounds present in *C. speciosus* have not been investigated against seasonal influenza. The leaf material of this plant is consumed raw or minimally processed; thus, we investigated the antiviral effect of the leaves rather than examining root specimens.

As a preliminary investigation, in this study, we investigated the TB100 time of addition effect and its effect on virucidal, viral attachment, and entry against the A (H1N1) infection in RAWA264.7 cells. We observed that TB100 had a profound pretreatment effect, but not as a co- or posttreatment candidate. In vitro studies demonstrated that the potency of TB100 pretreatment on RAW264.7 cells was comparable to that of IFN-β, a prominent marker and modulator in the antiviral defense of the body [42]. Thus, we speculated that TB100 is suitable for further investigations as a pretreatment candidate against influenza. When the cells were treated with TB100, microscopic observations revealed no significant cellular morphological alterations. During in vitro antiviral assays, a significant reduction in viral copy number was observed after 12 h of exposure to 20 and 40 µg/mL for RAW264.7 cells. These observations are comparable to those for IFN-β (PC). The antiviral effect of TB100 was not confounded by phagocytic cells but remained with non-phagocytic cell lines, such as A549 and HEp2. Immune stimulatory effects exerted on both phagocytic and non-phagocytic cells can be important for alleviating the disease under natural conditions, where both types of cells contribute to host protection as components of the innate immune system. To ascertain the ability of TB100 to mount an immune modulatory antiviral effect, we selected IFN-β and IL6 as two of the key markers of the antiviral defense of the body. Both IFN-β and IL6 exhibit pro- and anti-inflammatory properties [42]. These activities play essential roles in mounting an appropriate adaptive antiviral defense [43,44]. The exposure of RAW264.7 cells to TB100 caused significant induction of both IFN-β and IL6 cytokine levels in ELISA and qRT-PCR formats with just 24 h of exposure, indicating its potential to evoke adaptive antiviral signatures against viral infections. During a natural viral infection, viral antigens are sensed by a variety of cell surface and intracellular receptors and activate cellular kinases such as TBK1 and IKKε which subsequently induce transcriptional activators such as IRF3, IRF7, ultimately leading to an NF-kB-mediated cellular antiviral state. In this aspect, TB100 may assist the immune responses to bring them to a peak, subsequently leading to the activation of feedback loops that bring down the pro-inflammatory state [45].

The type I interferon pathway activation is an essential immune defense response against invading viral pathogens [46,47]. TBK1, IRF3, STAT1, IKB-α, and p65 transcriptional regulators are key elements in type I interferon signaling that regulates the nuclear localization of NF-kB (Figure 6), which directs the activation of at least 150 genes involved in defense responses [48]. The treatment of RAW264.7 cells with TB100 caused varying degrees of phosphorylation of these protein markers, potentially leading to transcriptional activation of antiviral genes. Using Western blot analysis and qRT-PCR, we observed a temporal effect on the phosphorylation levels of each of these protein markers. Significant dwindling phosphorylation responses were observed within 8 to 24 h of TB100 treatment. The maximum level of phosphorylation was observed after 8 h of treatment and reduced towards 24 h, possibly due to the holistic effects of the anti-inflammatory compounds present in TB100. There was profound antiviral activity even at the lower treatment concentration of 20 µg/mL, despite there being no significant induction of IFN-β or IL6 at that concentration at an early time point (12 h). This observation indirectly suggests other TB100 antiviral modes of action to its immune stimulation effect as a pretreatment. Creating an antiviral defense response, TBK1 (serine–threonine kinase) plays an essential role in regulating inflammatory responses to foreign agents. Following the activation of toll-like receptors by viral or bacterial components, TBK1 associates with TRAF3 and TANK and phosphorylates the interferon regulatory factors (IRFs) IRF3, IRF7, and DDX3X. This activity allows subsequent homodimerization and nuclear translocation of the IRFs leading to transcriptional activation of proinflammatory and antiviral genes including IFN-α and IFN-β [49]. IRF3 is a key transcriptional regulator of type I interferon-dependent immune responses and plays a key role in innate immune responses against DNA and RNA viral infection [50]. It is important in both early and late phases of immune responses against viral infections that involve indirect activation of IFN-α, IFN-β, and interferon-stimulated genes (ISG). Under normal conditions, IRF3 remains inactive in the cytoplasm; however, during viral infection, it is activated by phosphorylation, resulting in a major antiviral defense. STAT1 activates responses to interferons (IFNs), the cytokine KITLG/SCF, and other cytokines and growth factors. When type I interferons bind to their cellular receptors, they activate JAK and tyrosine phosphorylation of STAT1 and STAT2. Phosphorylated STATs dimerize and form a complex called the ISGF3 transcription factor that ultimately activates IFN-stimulated genes against viral infections [51], preventing the activity of dimeric NF-kB/REL complexes by trapping REL dimers in the cell cytoplasm. When a protein is phosphorylated, it is marked for ubiquitination and degradation, establishing the aforementioned interaction [52]. NFkB-p65 is a subunit of the NF-kappa-B transcription complex that plays a crucial role in biological processes such as inflammation, immunity, cell growth, differentiation, etc. Phosphorylation of p65 promotes the activation of NF-kB but otherwise remains inactive by complexing with IkB proteins [53]. Phosphorylation of these transcriptional activators by TB100 representing at least three major axes of antiviral pathways, such as the RIF1-IRF3/IRF7, TRIF-IKKa-p65, and JAK-STAT, indicates that the active ingredients present in TB100 might influence multiple immune modulatory pathways, necessitating further investigation (Figure 6).

The preservation of the antiviral activity of TB100 must be validated in vivo. To investigate the ideal dose and protective efficacy of various treatment concentrations, we first performed an in vivo study in the BALB/c mice that underwent influenza A (H1N1) infection in their lungs, demonstrating complications comparable to those observed in humans. Hence, the BALB/c mice model is suitable for dose optimization and drug efficacy assessments for influenza viruses [54]. Oral treatment of TB100 in the BALB/c mice at varying concentrations (1.05–8.40 mg/kg) did not cause significant adverse effects on body weight or behavioral changes. The mice were orally treated for four consecutive days (one day apart) to maximize the TB100 effect. If a drug is effective, the active compounds must be absorbed from the gut and metabolized to activate immune defenses. Upon challenge with influenza A (H1N1) lethal dose (twice the LD50), the mice that did not receive TB100 treatment were dramatically affected. By the 8th day after the challenge, all VO mice had a more than 20% lower body weight in comparison to those receiving the highest dose of TB100 (8.40 mg/kg; 0.8 mg/mL group). The treatment effect was evident as early as 3^rd^ DPI on histopathological examination and viral titers in the lung tissue. Toward the end of the experiment, the histopathological changes were more profound in the TB100- than in the VO-treated groups as the TB100-treated tissues regained normal (healthy) tissue architecture. Typical histopathological alterations due to influenza infections, such as high lung inflammation and exudate-filled lung spaces, could be observed in the VO group, whereas dose-dependent protection was evident in TB100-treated groups with almost normal lung architecture in the 0.8 mg/mL group [55]. TB100 activity was broad toward influenza strains other than A (H1N1). Here, we observed protective responses against A (H3N2) and (H9N2) influenza strains, resulting in immune augmentation against the viral activity. It was also noted that the mice showed greater resistance to influenza A (H3N2) and A (H9N2) strains than to A (H1N1), even at twice the LD50 nasal challenge. The first mortality for VO control was recorded in A (H1N1) as early as 6^th^ DPI and all were dead by 8^th^ DPI. However, the A (H3N2) and A (H9N2) VO groups never resulted in a 100% mortality. TB100 treatment demonstrated effectiveness against all three types of influenza. The protection obtained by TB100 oral treatment must be a holistic effect of the active compounds present in *C. speciosus* leaves; therefore, it is important to identify the compounds that may be responsible for its antiviral activity.

To identify these potential compounds, we conducted HPLC analyses. Among several candidates, we selected three major compounds, cinnamic, caffeic, and chlorogenic acids, as potential antiviral compounds. The availability of these compounds in the leaf extract was less than 0.2 % (*w*/*w*); however, their effect can provide significant antiviral immunity to the host. In vitro, antiviral assays conducted with all three compounds at varying 2, 10, 20, and 55 µM concentrations as a pretreatment demonstrated their significant ability to suppress the A (H1N1) viral activity in RAW264.7 cells. The determination of the EC50 and CC50 values for the three compounds demonstrated that cinnamic and caffeic acid had comparable outcomes, whereas the chlorogenic acid effect was more profound. According to Western blot assessment, the treatment of RAW264.7 cells with each compound induced the expression of transcriptional regulators and caused phosphorylation of those markers, indicating their propensity to induce antiviral defenses. Chemicals such as caffeic acid can interfere with viral neuraminidase activity [56]. Neuraminidase is an essential protein that sheds viral particles from infected cells. Here, we observed that all three compounds induced antiviral gene expression via phosphorylation and potentially suppressed inflammation by down-regulating inflammatory cytokines such as IL1β without compromising the antiviral effect. This phenomenon could potentially be important in treating viral infections, as virus-mediated cytokine storms are detrimental to patient health [57]. Despite the fact that we identified cinnamic, caffeic and chlorogenic acids as the three compounds that may confer antiviral activity of TB100, they may not accumulate; in addition, they metabolized in concentrations that we used in the in vivo study (8.4 mg/kg). However, they may confer antiviral activity even at minute concentration under in vivo conditions due to holistic effect of TB100 as herbal treatment. Owing to the richness of phytochemicals present in *C. speciosus*, its profound antiviral activity may be due to the holistic effect of numerous chemicals that we did not test for. Further studies are required to evaluate the biological effects of this plant on other viral diseases. To the best of our knowledge, this is the first study on *C. speciosus* leaf extract as a pretreatment against influenza, asserting its antiviral aspects for further investigation.

Activation of the interferon pathway against influenza infection (RNA viruses) could occur in multiple mechanisms. (1) IRF3, IRF7 axis, (2) p65 axis, and (3) JAK-STAT axis are some of the major mechanisms. Similar to viral infection, active compounds present in TB100 also potentially stimulate at least three proposed immune induction pathways that lead to interferon production and activation of NF-kB-targeted genes. Phosphorylation of representative markers in each of the pathways was evident upon cell treatment with TB100. The holistic activation of antiviral defense could be a cumulative effect of multiple active compounds present in TB100. 

## 5. Conclusions

The leaf extract of the medicinal herb *C. speciosus* can be an important source for the development of medical interventions to prevent seasonal health burdens such as influenza and may be useful during pandemic outbreaks. Furthermore, investigations directed at the identification of active compounds in other plant parts such as the bark, root, and flowers may have different active ingredient efficacies and profiles. This suggests that the genetic wealth hidden in plant resources could be a rich source of chemicals that hold the cure for influenza and many other future diseases faced by humanity.

## Figures and Tables

**Figure 1 viruses-15-01375-f001:**
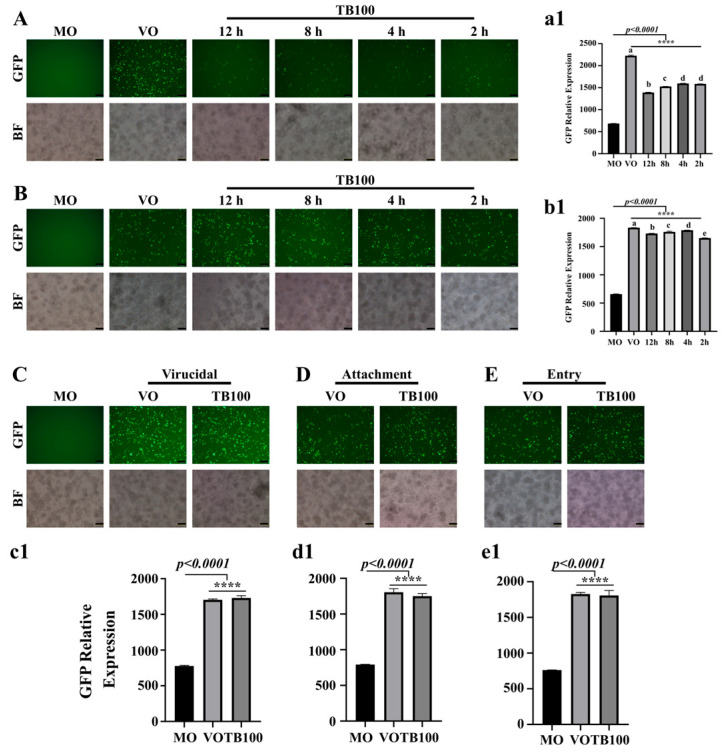
Time of addition, virucidal, attachment, and entry assays. (**A**) TB100 time of addition as pretreatment was investigated in RAW264.7 cells. Cell treatment with TB100 was conducted 12, 8, 4, and 2 h before infection with A (H1N1) PR8-GFP at MOI 0.1. After a 2 h incubation for infection, the medium was aspirated and complete medium was added. Cells were further incubated for 24 h. Viral replication was quantified using fluorescent imaging and GFP absorbance measurements (**a1**). (**B**) TB100 time of addition as posttreatment. RAW264.7 cells were treated for 2, 4, 8, and 12 h after the infection at MOI 0.1. After incubation for 24 h, viral replication was quantified by fluorescence imaging, GFP absorbance measurements (**b1**). (**C**) TB100 virucidal effect. The virucidal effect of TB100 was investigated by mixing TB100 and A (H1N1) PR8-GFP before adding them on to DMEM-pre-washed (3 times) RAW264.7 cells. Incubation was performed for 24 h and fluorescence imaging and GFP absorbance measurements were conducted (**c1**). (**D**) Attachment assay. TB100 effect on viral attachment was conducted by mixing TB100 with A (H1N1) PR8-GFP before infection. Incubation was conducted at 4 °C for 2 h. Infection medium was aspirated and incubated for 24 h. Fluorescence imaging and GFP absorbance measurements were conducted (**d1**). (**E**) Entry assay. TB100 effect on viral entry was investigated in RAW264.7 cells by treatment after 2 h of infection at 4 °C. After TB100 addition, cells were further incubated for 24 h and fluorescence imaging and GFP absorbance measurements were conducted (**e1**). All experiments were conducted at least three times. Stars indicate the significant difference against MO control. **** indicate significant difference against MO control. The level of significance was determined at *p* < 0.05. Different letters indicate significant mean difference among each condition (Tukey multiple comparison method). Mean ± SD of three independent experiments were demonstrated. MO: Media only, VO: Virus only, TB100: TB100 treatment. The scale bar represents 100 µm.

**Figure 2 viruses-15-01375-f002:**
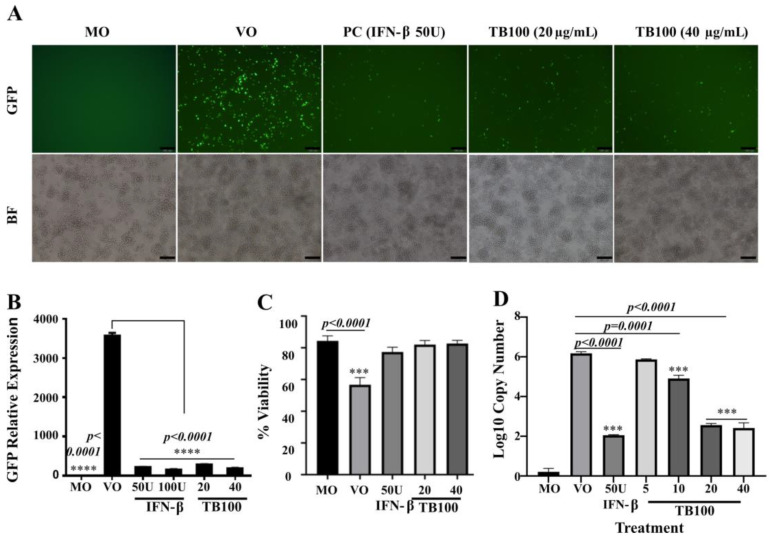
In vitro assessment of antiviral activity of TB100. (**A**) Fluorescent microscopy. Inhibition of influenza A (H1N1) PR8-GFP viral replication by TB100 pretreatment was investigated in RAW 264.7 cells. Dose-dependent viral suppression by TB100 was compared against the positive control IFN-β (50 U). (**B**) Fluorescent absorbance measurements. Quantitative measurements of viral suppression were measured in RAW264.7 cells after TB100 pretreatment by GFP absorbance. TB100 dose-dependent suppression was quantitatively investigated. **** indicate significant differences of treatments against the VO control. (**C**) Cell viability assay. The degree of cell protection by TB100 upon viral infection was carried out by cell viability assay using a hemocytometer (trypan blue exclusion method). Dose-dependent cell protection was investigated. A significant difference was compared against the MO control. (**D**) Reduction in viral copy number. Suppression of viral activity was quantified by qRT-PCR-based viral copy number determination. Significant differences were compared against the VO control. The level of significance was determined at *p* < 0.05. *** indicate significant difference against the MO control.MO: Media only, VO: Virus only, PC: Positive control. Means ± SD of three independent experiments were demonstrated. The scale bar represents 100 µm.

**Figure 3 viruses-15-01375-f003:**
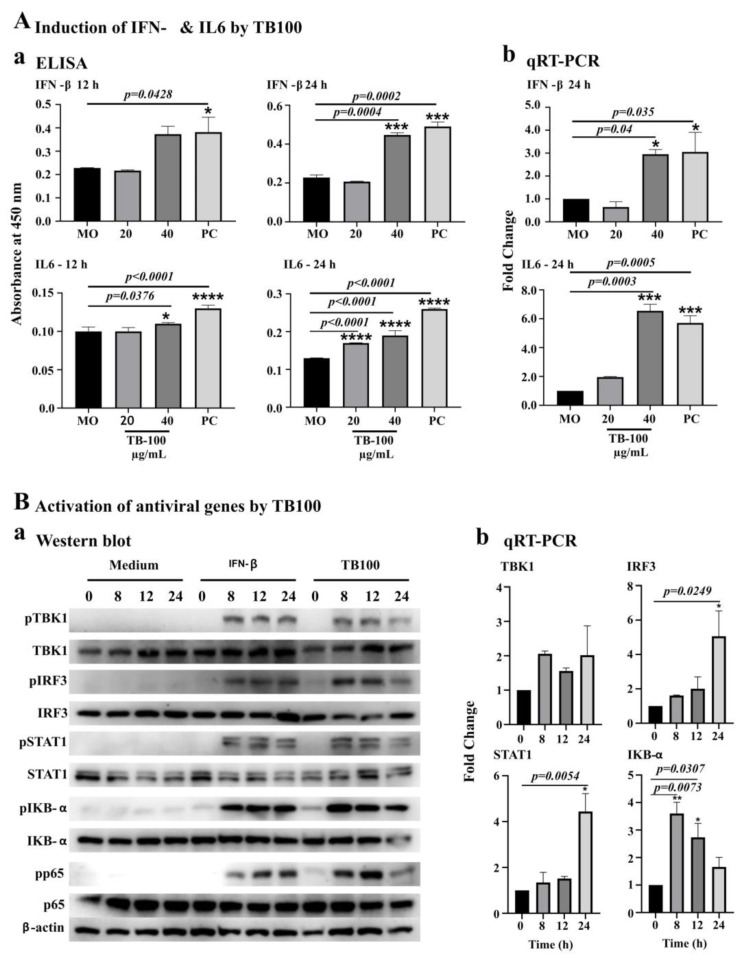
Induction of antiviral markers by TB100. (**A**) Induction of IFN-β and IL6 by ELISA. Induction of IFN-β and IL-6 antiviral markers was investigated in RAW264.7 cells after pretreatment with TB100. Dose-dependent activation was investigated 12 and 24 h after treatment. Quantitative assessment was conducted (**a**) via ELISA assay and (**b**) by qRT-PCR. Significant differences were compared against the media alone controlled by Tukey multiple comparison method. *, ***, **** indicate significant difference against the MO control. (**B**) Activation of antiviral proteins by phosphorylation. The RAW264.7 cells were treated with TB100 (40 µg/mL) and protein extraction was performed at 0, 8, 12, and 24 h after treatment. Level of phosphorylation of each protein marker was conducted using Western blot analysis (**a**). The expression of selected marker proteins was evaluated by qRT-PCR (**b**). Significant differences were compared against the zero-hour control. The level of significance was determined at *p* < 0.05. *, ** indicate significant difference against o h time point. MO: Media only, PC: Positive control. ELISA and qRT-PCR experiments were conducted as three independent trials. Mean ± SD is demonstrated.

**Figure 4 viruses-15-01375-f004:**
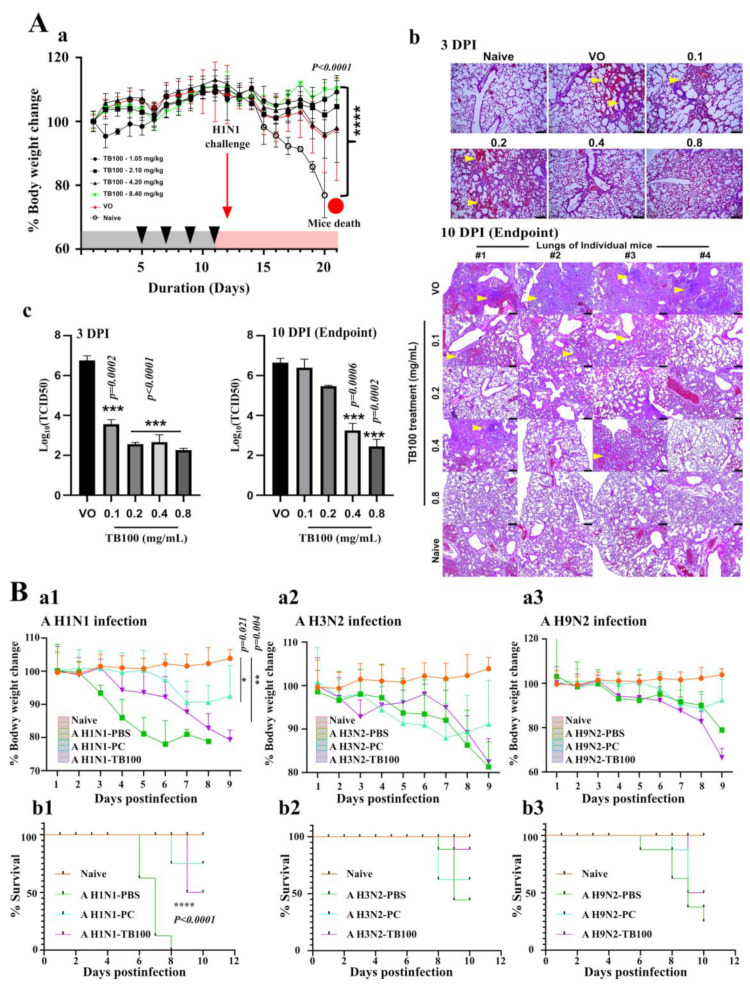
In vivo dose optimization of TB100 and multiple influenza A virus challenge study. (**A**) In vivo, a dose optimization study was conducted in the BALB/c mice model (*n* = 8, N = 48). Female BALB/c mice were orally fed with TB100 at a 0.1, 0.2, 0.4, and 0.8 mg/mL concentration on 5th, 7th, 9th, and 11th day and nasal challenge with A/Puerto Rico/8/1934 (H1N1) (double LD50) was conducted on 12th day. (**a**) Body weight measurements were conducted daily from day zero. Mean body weight ± SD is demonstrated. (**b**) Histopathological examination of lung tissues was conducted at 3^rd^ and 10^th^ DPI and was carried out by hematoxylin and eosin staining method. Arrows indicate sites of heavy inflammation. (**c**) Reduction in viral copy number was determined in lung tissues at 3^rd^ and 10^th^ DPI. Stars indicate significant differences against VO control. Mean TCID50 ± SD is demonstrated. The level of significance was determined at *p* < 0.05. (**B**) In vivo challenge study. BALB/c mice (*n* = 8, N = 56) orally treated with TB100 at 0.8 mg/mL four times in two-day intervals as for dose optimization. The mice challenge was conducted with influenza A (H1N1), A (H3N2), and A (H9N2) strains using double LD50. Post-challenge body weight measurements for (**a1**) A (H1N1), (**a2**) A (H3N2), and (**a3**) A (H9N2) were taken daily. Mean body weight ± SD is demonstrated. Statistical comparison was performed against the naïve group (unpaired t-test) at the end of the experiment. The mortality of mice was monitored and recorded on a daily basis. Kaplan–Meier survival curves were constructed: (**b1**) A (H1N1), (**b2**) A (H3N2), and (**b3**) A (H9N2). The statistical comparison was performed by log-rank (Mantel–Cox) test. The level of significance was determined at *p* < 0.05. VO: Virus only, PC: Positive control, DPI: Day post-infection. The scale bar represents 100 µm.

**Figure 5 viruses-15-01375-f005:**
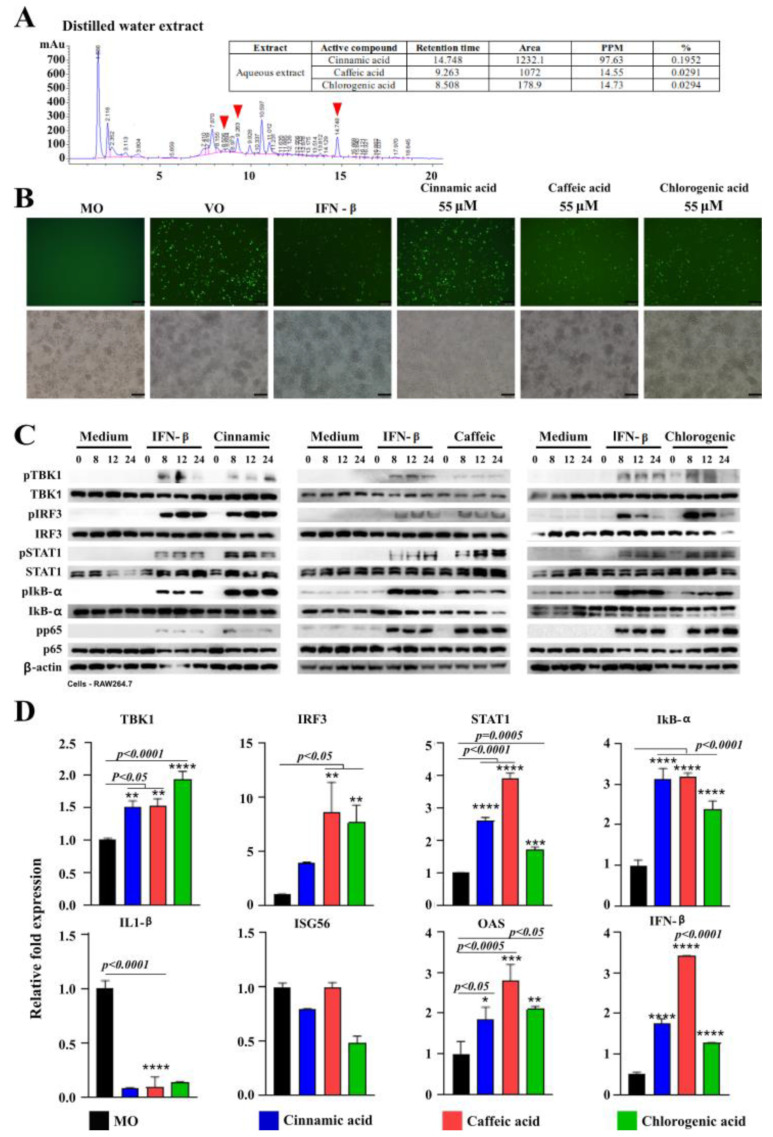
Identification of antiviral compounds of TB100 and assessment of antiviral activity. (**A**) HPLC analysis. HPLC was carried out to identify prominent compounds present in TB100 using distilled water and methanol-extracted samples. We picked up cinnamic, caffeic, and chlorogenic acids as major compounds present in TB100 using standard-based selection procedure. The antiviral activity was investigated in a dose-dependent manner. (**B**) GFP fluorescence imaging. Suppression of influenza A (H1N1) PR8-GFP in RAW 264.7 cells was investigated as pretreatment with each compound. Fluorescence imaging was conducted. (**C**) Western blots analysis. RAW 264.7 cells were pretreated with each compound (2 mM). Protein isolation was conducted in time intervals at 0, 8, 12, and 24 h after treatment. (**D**) qRT-PCR assessment of antiviral gene expression. The effect of cinnamic, caffeic, and chlorogenic on antiviral genes was investigated by qRT-PCR analysis. Statistical comparisons were conducted against the media-alone control (negative control). Three independent trials were undertaken and mean ± SD is demonstrated. The level of significance was determined at *p* < 0.05. *, **, ***, **** indicate significant difference against the negative control. Scale bar represents 100 µm. MO: Media only, VO: virus only.

**Figure 6 viruses-15-01375-f006:**
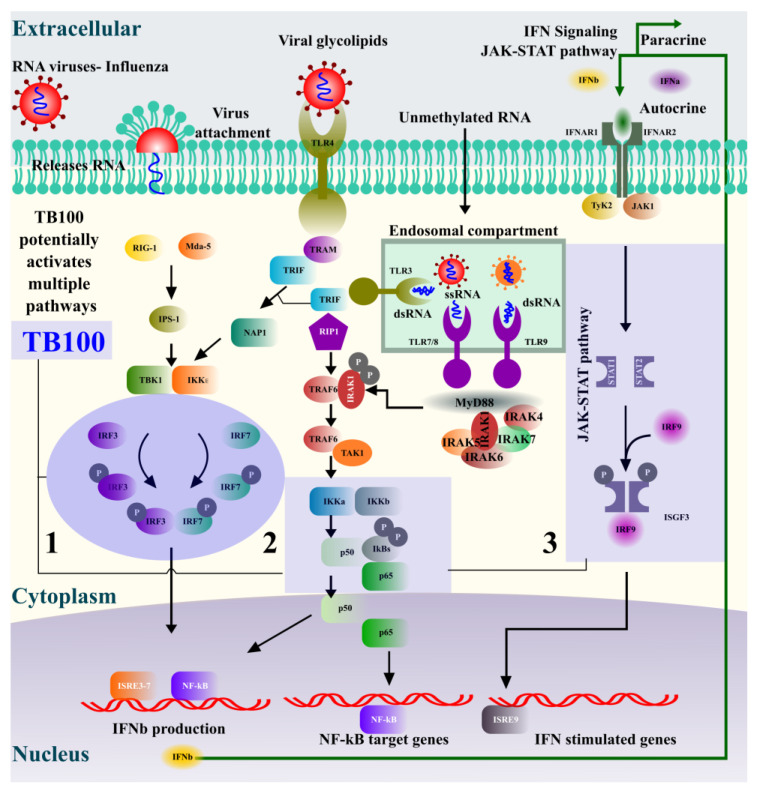
Potential mechanisms of TB100-mediated antiviral immune response induction.

**Table 1 viruses-15-01375-t001:** List of primers used in the study.

Target Gene	Primer Name	Sequence (5′-3′)	Ref./Gene Accession
TBK1 (Mouse)	TBK1 FP	CGGAAAGAAGTGTTGCGGTTAGC	[25]
	TBK1 RP	CAGGCTGCTTTTGCCATTGGTG	
IRF3 (Mouse)	IRF3 FP	CGGAAAGAAGTGTTGCGGTTAGC	[26]
	IRF3 RP	CAGGCTGCTTTTGCCATTGGTG	
STAT1 (Mouse)	STAT1 FP	GCCTCTCATTGTCACCGAAGAAC	NM_001357627
	STAT1 RP	TGGCTGACGTTGGAGATCACCA	
IKB-α (Mouse)	IKB-α FP	GACATGCCTCTCTCCTGTAGTC	[27]
	IKB-α RP	GGTGAAGCACATCACTGGTCTC	
IL-1β (Mouse)	IL1-β FP	GGTGTGTGACGTTCCCATTA	XM_006498795
	IL1-β RP	ATTGAGGTGGAGAGCTTTCAG	
ISG56 (Mouse)	ISG56 FP	AGAGAACAGCTACCACCTTT	AH004732
	ISG56 RP	TGGACCTGCTCTGAGATTCT	
OAS (Mouse)	OAS FP	GAGGCGGTTGGCTGAAGAGG	NR_126529.1
	OAS RP	GAGGAAGGCTGGCTGTGATTGG	
IFN-β (Mouse)	IFN-β FP	TCCAAGAAAGGACGAACATTCG	X14029
	IFN-β RP	TGAGGACATCTCCCACGTCAA	
GFP	GFP FP	GGCAAGCTGACCCTGAAGTT	ON755320
	GFP RP	CTTGTAGTTGCCGTCGTCCT	
GAPDH (Mouse)	GAPDH FP	AGGTCATCCCAGAGCTGAACG	NG_148718
	GAPDH RP	CACCCTGTTGCTGTAGCCGTAT	

**Table 2 viruses-15-01375-t002:** Determination of CC50, EC50, and SI values of TB100 in RAW264.7, A549, and HEp2 cells.

Cell line	CC50 (µg/mL)	EC50 (µg/mL)	Selectivity Index (SI)
RAW264.7	117.12 ± 18.31	15.19 ± 0.61	7.68 ± 0.89
A549	71.86 ± 8.91	13.64 ± 0.55	5.25 ± 0.44
HEp2	>300	16.78 ± 0.42	>17.87

**Table 3 viruses-15-01375-t003:** Assessment of CC50, EC50, and SI values for selected active compounds in TB100 using RAW264.7 cells.

Active compound	CC50 (mM)	EC50 (µM)	Selectivity Index (SI)
Cinnamic acid	6.076	28.64	212.150
Caffeic acid	8.925	28.18	316.713
Chlorogenic acid	4.875	15.62	312.099

## Data Availability

The data that supports the findings of this study are available from the corresponding author upon reasonable request.

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
