# Peer review of "The Aqueous Leaf Extract of the Medicinal Herb Costus speciosus Suppresses Influenza A H1N1 Viral Activity under In Vitro and In Vivo Conditions"

_viruses, 2023, doi:10.3390/v15061375_

Round 1

Reviewer 1 Report

1. Fig 1A and 4A, adjusting the white balance can improve the clarity of the bright field image captured by the fluorescent microscope.

2. Scale bars should be included in all images of fluorescent microscope and H&E staining, and their corresponding values should be clearly indicated in the figure legends.

3. Fig 1-4, it is uncommon to include the names of experiments in figures. To provide a clearer presentation of the data, the authors could remove this information from the figures and instead providing more comprehensive details of the experiments in the figure legends.

4. Fig 1-4, all statistical graph in this paper should clearly indicate the data sets that are being compared.

5. Fig 1-4, I do not think the ANOVA mentioned in the Materials and Methods is appropriate for analyzing all the data in this paper. The authors should provide an accurate description of the statistical analysis used for each figure in the respective legends.

6. Fig 3, survival curves lack statistical analysis.

7. Fig 2-3, to avoid confusion, would it not be simpler to label the figures as Fig 2A-D and Fig 3A-E instead? Currently, there seems to be a mislabeling issue as even the authors themselves mistakenly labeled Fig 3A-c and 3B-b as Fig 3A-d and 3B-c in the figure legends.

8. The well-known mechanism is that IFN-I activates the JAK-STAT pathway and provides feedback regulation to the IRF3 and NFKB pathways through ISGs. I am not sure whether IFN-I can directly and efferently induce the phosphorylation of TBK1, IRF3, and p65, please provide some references for clarification on this matter.

9. Fig 4A, can HPLC identify unknown compounds without mass spectrometry? Or were the prominent compounds detected in TB100 already known? Please provide more detail on this section.

10. Fig 4C and Table 1, the correct way to write IL1-β is IL-1β. 

Author Response

Responses to reviewer 1 file is attached 

Reviewer 2 Report

The manuscript illustrates the antiviral activity of an herb called Costus speciosus against influenza virus infection. In vitro results, Costus speciosus is a non-cytotoxic herb that can inhibited the viral replication dramatically through inducing interferon-related pathways. In vivo data descripted the Costus speciosus is a safe drug that effectively protected mice from influenza viruses infection.

The data provided do support the main conclusion of the authors, however, some open issues might be addressed from the authors.

(1) Fig.2A(a,b): the author showed that 20µg/mL drug could not induce the transcription and secretion of IFN-β at 12h and 24h. Can the authors explain why the 20µg/mL drug in Fig.1 showed significant antiviral activity?

(2) Fig.2B(c): data suggested that the level of interferon related genes phosphorylation at 24h was significantly lower than that at 8h and 12h after TB100 treatment. Why was there no difference in the secretion of IFNβ and the transcription level of IFNβ remained high (compared with LPS) in Fig.2A?

(3) Fig3A(a): What is Naive meant to indicate? why was there such a big difference in body weight between the VO group and the native group, but the pathological injury in lung from the Naive mice were more slight than the VO mice(Fig3A(b)), and the VO group was comparable to the TB100-4.20 mg/kg treatment group? 

(4) The Fig.1A and Fig.4A: these fluorescent results are very confusing. In the absence of viral infection (MO), this cell appears to be in a poor state, with cells aggregating into clumps? In addition, I observed that the two BF cell images in the PC panel and in TB100(40 µg/mL) panel appear to be one visual field, and the BF cell image in the PC panel is the inversion of the TB100(40 µg/mL) panel. Can you please provide the raw data?

Minor issues:

1. It is not indicated which PR8-GFP variant was used. Please provide a reference.

2. The antiviral genes is clearly induced upon TB100 treatment, but the author still need to confirm it under influenza viruses infection.

3. Fig. 2B(c): the western blot data should be quantified.

4. Fig. 3b: Criteria for sacrifice of animals should be proved in legend. And the difference significance analysis should be also done.

Author Response

Responses to reviewer 2 file is attached.

Reviewer 3 Report

The authors report the results of in vitro and in vivo anti-influenza A virus studies with an aqueous leaf extract of Costus speciosus. In addition, the authors state that a methanol extract exerts similar activity in vitro.  

To make the paper more complete the following comments have to be addressed and some missing points have to be included:

The title is very long. I suggest to delete “broad-spectrum efficacy” because the antiviral effect on influenza B viruses was not addressed in the study.

The introduction summarises the medical impact of influenza virus infections, treatment options, and the potential of herb preparations. However, the introduction does not summarize well the knowledge of biological activities and ingredients/compounds of aqueous leaf extract of Costus speciosus (TB100). Please explain why the anti-influenza virus activity of leaf and not root extract was studied.

The material and method section has to be written more carefully.

-        -  Please describe in all paragraphs of the material and method section how often the experiments were performed and how many parallel measurements provide the basis for the mean values and SD shown.

-        -  Please start the material and method as well as the results section with the evaluation of cytotoxicity and describe that non-cytotoxic concentrations were applied in the following antiviral studies.

-       -   Next, please describe the EC50 determination the material and method as well as the results. Why only the inhibition of virus replication after pretreatment with TB100 was studied if nothing was known about the anti-influenza virus activity of the leaf extract? Generally, the CC5 and EC50 values will help the reader to understand the selection of concentrations for the following studies.

-     -     The authors state in the introduction that there are no data on the anti-influenza virus effect of aqueous leaf extract of Costus speciosus. Therefore, it will be important to generate and add data on the antiviral effect of TB100 when added (i) during virus adsorption, (ii) after adsorption and (iii) during the whole experimental time. Please include an antiviral drugs as positive control in these studies. The results could help to explain why the authors were focusing on the antiviral effect of pre-treatment in their study.

-        -  The title of 2.4 and 2.5 is misleading because the authors assessed only the antiviral effect of pre-treatment with TB100 in these paragraphs.

-         - 2.11: Please explain the selection of time points for treatment.

-        -  2.12: Please describe the source of viruses applied in vivo.

-       -   2.13: Plant extract often contain polyphenols known to inhibit virus adsorption. The authors should check the content of polyphenols of TB100 and perform a control experiment to check the antiviral activity after removal of polyphenols.

Generally, the results demonstrate that pre-treatment with TB100 induces an effective interferon type I response and modulates antiviral signalling which might explain the described antiviral activities of TB100 in vitro (at least in RAW265.7 cells) and in vivo.

No data were presented for other treatments (during virus adsorption, after adsorption, and during the whole experimental time). To significantly improve the manuscript the authors should perform a time-of-addition experiment prior to publication.

Fig. 1A, 3Ba, and 4A: The quality needs improvement.

Table 2: Please describe that the CC50 of TB100 in HepG cells was >300 µg/ml and the SI >17.87.

According to the authors, the content of cinnamic, caffeic, and chlorogenic acid in TB100 was less than 0.2%. I am wondering whether the systemic concentration of these compounds after oral administration of 8.4. mg/kg reached the described EC50 values. This should be addressed in discussion section.

Please adapt the abstract accordingly.

Author Response

Responses to reviewer 3 file is attached. 

Round 2

Reviewer 2 Report

  • The author's answer basically solved my concerns.

Author Response

Responses to reviewer is attached. 

Reviewer 3 Report

The authors addressed most but not all critical points mentioned by the reviewers. The introduction is much better now. The description of methods has been significantly improved and the results are better readable presented.

However, the following comments need to be addressed:

1.      Please correct the name of all influenza viruses according to the current WHO nomenclature (WHO. A revision of the system of nomenclature for influenza viruses: a WHO Memorandum. Bulletin of the World Health Organization, 58 (4): 585-591 (1980). Available at: https://apps.who.int/iris/handle/10665/262025) throughout the manuscript. For example, please replace in L18-19 “influenza A H1N1 PR8 virus” by “influenza virus A/Puerto Rico/8/34 (H1N1)”.

2.      All abbreviations should be introduced before using e.g. L19 EC50 and CC50,

3.      Paragraph 2.3: Please describe also the medium used in the experiments with virus infection.

4.      Paragraph 2.4: Please indicate the number of independent experiments and the parallel measurements. Please name of the statistical method in L146.

5.      Paragraph 2.5, L165: Please indicate the number of parallel measurements per time point per experiment.

6.      Paragraph 2.6: Please indicate also the number of parallel measurements per experiment.

7.      Paragraph 2.7: Please indicate the also number of parallel measurements per concentration.

8.      Paragraph 2.8: Please describe the method of EC50 determination soon after CC50 determination as paragraph 2.5 to have the same order as in the results section. Please give the number of independent experiments and of parallel measurements per concentration per experiment.

9.      Paragraph 2.9: Please indicate the number of parallel measurements per concentration per experiment.

10.  Paragraph 2.13, L329: Please cite literature example.

11.  The cytotoxicity experiments with cinnamic, caffeic, and chlorogenic acids needs to be described too e.g. together with the antiviral activity in paragraph 2.14.

12.  L334: should be “cinnamic, caffeic, and chlorogenic”.

13.  L362: Please add “CC50” to the paragraph headline -> “Determination of CC50, EC50, and selectivity index of TB100”

14.  Quality of the BF photographs in Figure 1, 2, and 5 is poor.

15.  Figure 2B: Was the difference of GFP relative expression between VO and treatment with IFN-ß or TB100 not significant?

16.  Figure 4A: According to Figure Aa all virus-infected control mice were killed on day 10 p.i. In contrast, the all treated mice survived the influenza virus infection. I am wondering how the authors determined the virus titres on day 10 p.i. for all experimental groups as shown in Figure 4Ac. Moreover, according to in Figure 4Ac there is no reduction in virus titre in the virus control group compared to day 3 p.i. This is very surprising. Usually, influenza viruses are eliminated by the adaptive immune response at that time point.

17.  Figure 4Ba1-3 is hard to understand. Please show the body weight changes in the same manner as in Figure 4Aa.

18.  Figure 5A: Due to the small letter size, this figure part is unreadable.

19.  L770: Please add that IL-6 is also a proinflammatory cytokine.

Author Response

Responses to reviewer is attached. 
